# Performance evaluation of therapeutic feeding centers through efficiency, effectiveness and utilization of bed capacity: A retrospective quantitative study

Ahmed Alhidary[1,2]*, Pesigan Arturo[1], Ali Al-Waleedi[3], Ferima Coulibaly-Zerbo[1], Omar Faisal[1], Mahammad Al Mansour[4], Ali AL-Mudwahi[4], Mohammed Rajamanar[4], Ezechiel Bisalinkumi[1], Eshrak Al-Falahi[1], Latifah Ali[1]

**1** EM/ACO/YEM/WHO Country Office, Sana'a,Yemen, **2** Department of Community Medicine, Faculty of Medicine, University of Aden, Aden, Yemen, **3** Ministry of Public Health and Population, Sana'a, Yemen, **4** Ministry of Public Health and Population, Aden, Yemen

* alhidary73@yahoo.com

## Abstract

The balance between wise use resources and provision of health services are crucial for the sustainability of services and ensuring value for money. Thus, it is essential to regularly monitor the status of health services to address any variations that arise. This study aimed to evaluate the performance of Therapeutic Feeding Centers (TFCs) by examining their efficiency, effectiveness, and beds utilization and assessing related factors. A retrospective quantitative study was conducted in 94 TFCs in Yemen from January to December 2022. Clinical outcomes like recovery rates, death rate, and default rate were measured, alongside with the dependent variables of efficiency, bed utilization, and effectiveness as a matrix to evaluate the TFC performance. The Data Envelopment Analysis model was used to measure technical efficiency, the Pabon Lasso model was used to classify the performance levels. The regression coefficient was used to determine associated factors. A total of 26,887 records of Sever-Acuate-Malnutrition children were reviewed. The results indicated that the recovered rate 90%, default rate 3.6%, death rate 1. 3%. The average effectiveness scores were 95%, technical efficiency 73%, and bed utilization rate 63%, the average performance was 82%. Moreover, the study found a positive weak correlation between effectiveness and efficiency (r = .190, n = 94, p = .008), and the efficiency was influenced by the Occupancy Bed Rate, Bed Turnover Rate and Length of Stay, $F_{(3,90)}$ =123.90, P < 0.001, $R^2$ = .805. The results indicate that the TFCs performed well overall and even exceeding Sphere standards thresholds for the outcome and effectiveness indicators. However, there is still a need to improve efficiency and bed utilization to meet the standards. The study indicated to a positive correlation between effectiveness and efficiency, although no relationship has been found between them. To further performance enhancement, the ministry of health and its partners must optimize resources utilization.

**Data availability statement:** All relevant data are within the manuscript and its Supporting Information.

**Funding:** The author(s) received no specific funding for this work.

**Competing interests:** The authors have declared that no competing exist

**Abbreviations:** TFC, Therapeutic Feeding Center; OTP, Outpatient Therapeutic Programs; SAM, Severe Acute Malnutrition; SAM/MC, Severe Acute Malnutrition with Medical Complications; IMAM, Integrated Management of Acute Malnutrition; CMAM, Community-Based Management of Acute Malnutrition; HFs, Health Facilities; BOR, Bed Occupancy Rate; BTO, Bed Turnover; LOS, Length of Stay; TE, technical efficiency; DEAP, Data Envelopment Analysis Program; DQA, Data Quality Assurance.

## Introduction

Severe Acute Malnutrition (SAM) is one of the most life-threatening problems facing children in developing countries. It is seen as one of the common causes of morbidity and mortality in children under the age of five [1]. Worldwide, about 17 million children under five suffered from SAM in 2018, and most of them were from the developing countries [2]. This serious situation puts children at high mortality risk since it raises the levels of severity and susceptibility of life-threatening infections and slows recovery [3]. SAM is causing more than one in ten deaths of children under five in the developing countries [4]. Malnutrition is not just a health issue; its prevalence among children mainly driven by inadequate dietary intake, frequent infections, and economic hardships [5] It is also associated with high health care expenses, low productivity, and slow economic growth, which can perpetuate a cycle of poverty and poor health [6,7]. It is therefore regarded a major global priority for states and organizations to pay attention to, as so indicated by the Sustainable Development Goals [8].

In Yemen malnutrition is not a new problem. It has been reported as an outcome of the current war. But its occurrence as a problem has been reported for many years now [9]. The situation has been aggravated by the war and has left its impacts on the humanitarian and economic situations in the country and since then has become a complicated problem. United Nations (agencies have frequently warned that Yemen is facing high food insecurity and malnutrition [10,11]. An estimated 2. 2 million children under the age of five suffer from under-nutrition, of whom 538,000 being severely malnourished [12]. Yemen Nutrition Cluster has reported that the Outpatient Therapeutic Programs (OTPs) received about 346,512 SAM children in 2021, while the OTP clinics received 387,540 SAM children by 2022, with a 12% increase, at a 90% reporting rate. Similarly, the therapeutic feeding centers (TFCs) received 32,436 children with severe acute malnutrition with medical complications (SAM/MC) in 2021 compared to 41,784 SAM children in 2022. This shows a 29% increase, thus exceeding the 2021 percentage with a 91% reporting rate [13].

SAM can be effectively managed in the community if devoid of medical complications. This is because complicated cases require inpatient management that provides admission to the TFCs [14], which provide inpatient care for children with SAM and medical complications. These centers offer 24-hour monitoring, therapeutic feeding, and treatment of infections and other conditions. TFCs are part of the integrated management of acute malnutrition (IMAM) approach, which also includes community-based management of acute malnutrition (CMAM) for children with SAM without complications. in 2022 161 of these TFCs across the country were run by the MoPHP in Aden and Sana'a TFCs with the support of various donors. The TFCs offer medical care and nutrition support to the SAM/MC children and provide education programs to their parents to protect their children from getting malnutrition and avoid relapsing of the recovered children [15].

Nutrition partners have invested many resources in setting up and running the TFCs so that they become accessible to all children in need. They also developed some guidelines in addition to establishing more TFCs in the affected areas. Some

applications monitored activities and measured monthly outputs, but they were not comprehensive enough to assess the TFCs in depth. There is need, for example, to measure how well the TFCs are functioning, how effective and efficient they are, and how well they utilize their capacity. This is crucial to lead and guide health planners and policymakers to design appropriate strategies and interventions to enhance the nutrition of malnourished children. Such steps are especially urgent because of the rapid increase of SAM cases every year. The TFCs are facing resource constraints while donor funding is in the decline.

The complexity of medical care provision, including the continued trend towards value for money and payment for services, is necessary in the health services industry. It represents the extent to which set objectives are being achieved [16]. It also reflects the physical relation between resource inputs and the health results achieved[17]. As a result, performance measurement of Health Facilities (HFs) is crucial for authorities, partners, and the society. It is a complex and multi-dimensional concept that includes assessing how effectively each health facility is providing medical care to its patients, and how optimally it utilizes its resources and capacity. The international agencies and academic researchers have proposed several sets of methods and models to help in assessing the performance levels. Some institutions and researchers have adopted a three-dimensional matrix to measure the performance of HFs as a quantitative method, covering the following indicators: efficiency, technical effectiveness, and bed utilization capacity [18]. Others have adopted only effectiveness and efficiency to measure performance [19], while other researchers used bed occupancy rate (BOR), bed turnover (BTO), and length of stay (LOS) to measure performance [20]. This proves the healthcare sector still needs the development of a clear and comprehensive model for evaluating and measuring performance so as to yield useful results [21].

Measuring the TFCs performance is essential in order to comprehend operational status, allocate resources in the most optimal way, and to guarantee transparency and accountability. This is especially necessary since the TFCs are getting support and funding from sources which are needed to verify everything is done properly. To the best of the team's knowledge, there is no study that has investigated performance evaluation of inpatient programs in a quantitative manner either locally or regionally. Research activities are not prioritized in countries with high prevalence of SAM/MC. In comparison to neighboring and other nations, scientific research activities in Arab countries are among the lowest globally. [22,23]. We therefore hope this study will add an important contribution to this issue in proposing the use of a three-dimensional quantitative approach (effectiveness, efficiency, and bed utilization) to apprize the performance level of inpatient medical care. The approach itself will help policy makers to detect weaknesses in hospitals systems and support the process of their reform. Furthermore, the Pabon Lasso model has been used in a special manner in which the model is used to classify the performance level rather than the classification of efficiency of bed utilization. At the level of country, the study will provide reference to apply DEAP the teams as a technique to evaluate the efficiency and Pabon Lasso model to classify the performance level. In addition, the study will provide clarification for the TFCs situation and their performance as well as a reference value for the main indicators of the bed utilization rate in Yemen. The study seeks to answer three main questions: How well do the TFCs perform technically in managing the SAM/MC children? How efficiently do the TFCs use their resources and capacity to achieve their objectives? and what are the factors related to these indicators?

## Methods

### Study design

A retrospective review study of the TFCs records was conducted in 94 therapeutic feeding centers (TFCs) across 22 governorates for the period January to December 2022.

### Study population and setting

The data for this study came from 94 (67%) of the TFCs that were active in 2022 and supported by WHO, out of 141 TFCs allover Yemen. Of these 94 TFCs, 62 (66%) were in the northern part of Yemen, and 32 (34%) were in the southern part. The distribution of TFCs was based on the population density and prevalence of SAM in Yemen, which enhanced the

external validity of the study. The study covers all TFCs that have received WHO support. Centers that stopped functioning before the end of the study year were excluded from this study.

## Data quality assurance (DQA)

DQA is a process of verifying the data gathering and utilization by following the proper standards to ensure the data are valid, reliable, accurate, consistent, integral, and complete. In this study, a data collection method was applied that consisted of: 1) unifying and defining the performance indicators, 2) developing valid and reliable tools to collect data, 3) choosing the methods of data collection, 4) training the data collectors, 5) determining the sources of data collection, 6) cross-checking with other data sources, and 7) compiling and final reviewing, filtering the data at the central level.

## Evaluation and measurement technique

The study seeks to evaluate the technical performance of the TFCs by using a quantitative method to measure the effectiveness, technical efficiency, and bed utilization, based on the data of inputs, outputs, and intermediate outcome. Firstly, a technical effectiveness matrix has been built for the intermediate outcome variables according to the concept of the measurement effectiveness theory. This theory defines the effectiveness measurement as the deviation distance from a given state to some reference system state [24].

Based on several studies previously made, we have selected six intermediate outcome variables to determine the average for the effectiveness score, which include the recovered rate, readmission rate, relapsed rate, default rate, death ratio, referral to other TFCs rate and length of stay (LOS) rate. The cure indicator is summation of both cure and recovery cases. It was estimated by dividing the number of recovered children to those who were discharged. The other four indicators are calculated by subtraction the indicator value minus one divided by the total number of discharged children and multiplying them by 100% (applying a reversible formula (1- (indicator value/ discharge)) *100%). The LOS is a reference value of 7 days in the effectiveness matrix. We calculated the percentage of deviation from the reference value to the actual value, by using the formula (1- (absolute value (7-LOS)/7)) *100). Thus, the LOS of 7 days received 100%, and the LOS of 6 or 8 days received 86%…etc. Then, estimated the utilization of the bed capacity matrix from the OBR, the BTO and the LOS.

Calculating the efficiency by estimating the technical efficiency (TE) by using the DEAP Ver 2.1 program to conduct Data Envelopment Analysis (DEAP). DEAP is a technique that evaluates the relative efficiency of a set of units that use the same inputs to produce the same outputs [25,26]. In this article, the input variables included the number of staff, number of active beds, amount of staff incentives, and operational cost payment, while the output indicator is based on the number of admitted children.

## Data analysis strategy

The data has been collected from the TFCs records on a regular basis by using a valid and reliable checklist. The checklist included two parts, the admission component, that includes remaining patients, transfer in (from other TFCs or OTP), re-admission, relapsed, new diagnosed and total inpatient days. Moreover, the discharge component which includes the recovered children, non-response cases, transfer out, defaulted cases and death cases.

1. A table was created containing twelve variables, which were displayed in a table before being manipulated and integrated into the assigned matrix for the study variables, Table 1.

2. Seven variables are chosen to construct the technical effectiveness' Matrix. Five of these variables are reversed, using the reversible formula, where two are calculated as percentages. These variables are estimated separately, then combined, and averaged to determine the technical effectiveness score. Bed utilization matrix has been built, by estimating three variables separately, the percentage of the OBR, the LOS, and the BTO, then combined and averaging them to

**Table 1. Summary of the inputs, outputs, and intermediate outcomes in the 94 TFC centers.**

| Indicators | Percentage per month | Total | Mean | SD | Minimum | Maximum |
|---|---|---|---|---|---|---|
| TFC Beds | – | 988 | 10.5 | 5.6 | 4 | 34 |
| Admission | – | 26,887 | 286 | 226 | 16 | 1,230 |
| TFC Staff | – | 1,516 | 16 | 5.7 | 7 | 41 |
| Readmission rate | 0.5% | 129 | 1.4 | 2.7 | 0 | 17 |
| Relapsed rate | 2.2% | 577 | 6.14 | 9.1 | 0 | 46 |
| Default rate | 3.6% | 944 | 10 | 15.2 | 0 | 94 |
| Death ratio | 1.3% | 356 | 3.79 | 8.7 | 0 | 60 |
| Recovered rate [b] | 90% | 23,782 | 253 | 195 | 15 | 1,141 |
| Transfer out [c] | 0.9% | 239 | 2.5 | 5.3 | 0 | 30 |
| OBR | 48% | – | 48% | 18.7% | 4% | 93% |
| LOS | 7.1 | – | 7.1 | 1.5 | 4 | 12 |
| BTO | 2.5 | – | 2.1 | 0.8 | 0.2 | 4 |

a Default rate absent for two consecutive days b. Recovered rate represents the proportion of children reached the discharge criteria from SAM treatment (cured from malnutrition). c. Transfer out moved to another health facility for further medical care or moved to receive medical care in another therapeutic feeding center

 get one value for the bed utilization indicator. The technical efficiency is estimated by using the DEA program. The set of inputs for the units consisted of staff size, active bed count, meal provision, and operational cost, while the set of the output is based on the number of sever malnourished children who received treatment in the selected TFCs.

3. The technical performance of the TFCs is calculated based on the quantitative method that averages the three indicators: technical efficiency (25%), bed utilization (25%), and technical effectiveness (50%). The performance results are classified into four groups: high result (80% - 100%), moderate result (60% - 79%), low result (> 59%). The Pabon Lasso model is used to classify the performance scores into four zones: Zone III (high effectiveness, high efficiency, and high bed utilization), Zone II (high effectiveness, high efficiency, and moderate bed utilization), Zone IV (high effectiveness, moderate efficiency, and moderate bed utilization), and Zone I (high effectiveness, low efficiency, and low bed utilization). Lastly, the average score is compared between the four zones, Table 3.

4. Finally, we have measured the significant relationships between the indicators and evaluated a significant effect of the variables on the main indicators.

## Conceptual framework of the study

Number of studies have adopted a three-dimensional matrix to measure the performance of health facilities, they focusing to quantitatively assess performance, focusing on indicators such as efficiency, technical effectiveness, and bed utilization capacity [27,28]. Others have addressed only effectiveness and efficiency to measure performance [29,30], while other researchers used bed occupancy rate (BOR), bed turnover (BTO), and length of stay (LOS) to measure performance [31]. In this study, the authors developed a conceptual framework to measure the TFC center's performance through a three-dimensional quantitative approach (effectiveness, efficiency, and bed utilization). This conceptual framework based on the Pabon Lasso Model. The Pabon Lasso Model is practical and valid tool to evaluate health facility in terms of effectiveness, efficiency, and bed utilization. It helps the facilities management and policymakers identify areas of defect and make informed decisions to optimize resource use. The conceptual framework of the study is presented in Fig 1.

 The framework integrates a three-dimensional quantitative approach—effectiveness, efficiency, and bed utilization—based on the Pabon Lasso Model to evaluate technical performance and guide resource optimization.

## Data analysis methods

This study aims to evaluate the performance of the TFCs by examining their efficiency, technical effectiveness, and beds utilization rate in addition to assessing any other related factors. The following statistical analysis methods are used to address this objective. The analysis of the data is done by using SPSS version 26 and Microsoft Excel version 2016. The mean, standard deviation (SD), minimum and maximum value and total of beneficiaries are computed for all variables. Then, the main indicators are summarized following the same statistics. The correlation analysis is used to explore the association between the indicators. The multiple linear regression model is also used to measure the impact of some variables on the three main indicators. The correlation coefficient (r) is used to determine the direction and strength of linear relationships and the regression coefficient determination ($R^2$), stepwise method, is used to assess the effect of variables on the main indicators. The F-test is used to evaluate the suitability of the regression model to the data, the significance level was set for all tests.

## Ethical considerations

The study utilized secondary data obtained from the WHO's monitoring and evaluation unit, adhering to all relevant ethical guidelines and regulations. The data were anonymized to prevent the disclosure of any personally identifiable information. The Ministry of Health granted permission to use the data, and four ministry employees co-authored the article. Efforts were made to ensure data accuracy, and all potential risks associated with data use were assessed to minimize harm to individuals or groups.

## Results

### Descriptive analysis

This study aims to evaluate the performance of the TFCs by examining its efficiency, technical effectiveness, and beds utilization rate and assessing related factors. Therefore, the study questions were: 1) what is the level of technical efficiency,

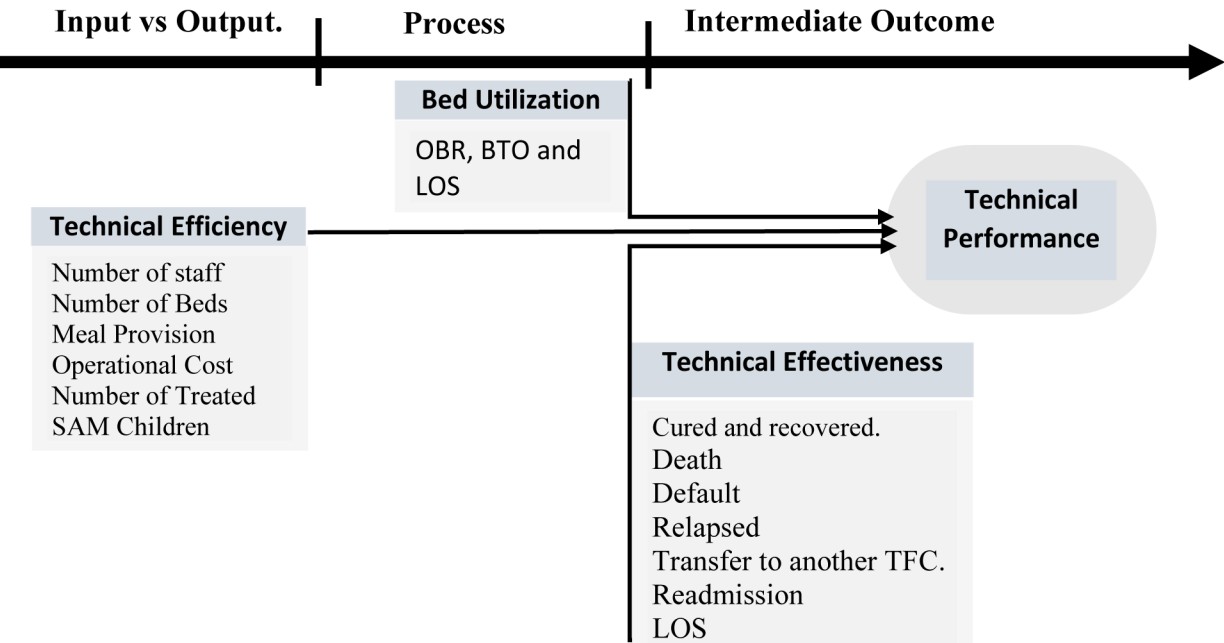

**Fig 1. A conceptual framework for measuring the TFC technical performance.**

technical effectiveness, and beds utilization rate of the TFCs and estimated the technical performance of the TFCs. 2) what are the factors related to these indicators? We conducted a retrospective review study of the TFCs records for 94 therapeutic feeding centers (TFCs) across 22 governorates for the period from January to December 2022. We created a matrix of output and intermediate outcome to measure the effectiveness and bed utilization rate. We used the DEAP Ver 2.1 program to measure the technical efficiency by using the same inputs to produce the same outputs of the TFCs. After that, the technical performance had been calculated by using an average of the three indicators. As can be seen from Table 1, the total active TFCs Beds is 988 beds in the 94 TFCs. Out of the 26,887 children who received treatment for malnutrition in these centers in 2022, 91% or 23,782 of them have recovered, while 1.3% or 356 of them have died, 3.6% or 944 children left the medical care before it was completed. The average length of stay in the TFCs was 7.1 with 48% bed occupancy rate.

## The technical performance estimation

To construct the effectiveness matrix, we collected seven variables using SPSS and Microsoft Excel programs. We applied the reversible formula to five of these variables and computed the percentages for the other two. We also estimated the bed utilization rate by taking the average of the BOR and BTO variables for each TFC center which we measured separately. Furthermore, we used the DEA program to measure technical efficiency. Lastly, we calculated the technical performance by averaging the three indicators. As can be seen from Table 2, the results showed that the technical effectiveness scored a very high 95%, with a very small standard deviation of 0. 03%. Moreover, the results indicate that the average bed utilization was 57%, with a standard deviation of 14%. The technical efficiency was also calculated. It reached an average value of 73% with a standard deviation of 14.37%. Finally, calculating the technical performance indicated that the average value of the performance was 82% with an SD of.07%.

Table 3 shows the performance of 94 TFCs in comparing between Pabon model results and the average of the performance scores. The Pabon model has classified 7 out of the 94 TFCs as entirely high performance (Zone III), while the average of performance scores revealed 63 TFCs to be high performance scores. As such, 15 TFCs, based on the Pabon

Table 2. A matrix to estimate the technical performance based on the bed utilization, efficiency, and effectiveness for the 94 TFCs.

| Indicators | Variables | Mean | SD | Minimum | Maximum |
|---|---|---|---|---|---|
| **Effectiveness** | Adjusted readmission value | 99% | .012% | 91% | 100% |
| | Adjusted relapsed value | 98% | .024% | 86% | 100% |
| | Adjusted default value | 97% | .40% | 84% | 100% |
| | Adjusted death value | 99% | .014% | 94% | 100% |
| | Recovered rate | 91% | 7% | 71% | 100% |
| | Adjusted transfer value | 99% | .02% | 90% | 100% |
| | LOS value | 83% | 14% | 29% | 100% |
| | Average rate | 95% | .03% | 88% | 100% |
| **Bed utilization** | OBR | 48% | 18.7% | 4% | 93% |
| | BTO value | 49% | 20% | 4% | 100% |
| | LOS value | 83% | 14% | 29% | 100% |
| | Average rate | 63% | 14% | 35% | 95% |
| **Efficiency** | Technical efficiency rate | 73% | 14.4% | 45% | 100% |
| **TFC Performance/ Average rate** | | 82% | .07% | 66% | 97% |

[a]for the variables readmission, relapsed, default, death and transfer, we reversed the score by subtract the 1 from the obtained result to get adjusted score. [b] for the variable BTO, we divided the obtained value by 4 to determine its percentage, and for the LOS, we divided the obtained value by 7 to determine its percentage. The numbers 4 and 7 are the average value for those variables in Yemen.

**Table 3. Comparison of TFCs performance results by using bed utilization, technical efficiency, and effectiveness/ average scores versus Pabon Lasso Results.**

| Rating of Indicators Score | | High [a] | Moderate [b] | Low [c] | Very Low [d] |
|---|---|---|---|---|---|
| Bed utilization | Number of HFs | 14 | 45 | 31 | 4 |
| | Percentage | 15% | 48% | 33% | 4% |
| Technical Efficiency | Number of HFs | 30 | 49 | 15 | 0 |
| | Percentage | 31.9% | 52.1% | 16% | 0 |
| Effectiveness | Number of HFs | 100% | 0 | 0 | 0 |
| | Percentage | 94 | 0 | 0 | 0 |
| Performance | Based on Average/ HFs (%) | 63 (67%) | 31 (33%) | 0 | 0 |
| | Pabon Lasso Results/ HFs (%) | 7 (8%) | 30 (32%) | 42 (44%) | 15 (16%) |

[a]High level ranges are above 80%, [b] moderate level ranges from 60% to 79%, [c] Low level ranges from 40% to 59%, [d] very low level is below 39%

model, were completely ineffectively performance (Zone I), the average estimation of performance did not detect TFCs were ineffectively performance. Similarly, the Pabon model showed that there were 42 in Zone IV, while the average estimation did not detect TFCs were in low performance. Finally, the Pabon Lasso model showed 30 TFCs were in moderate performance (Zone II), like the average score revealed 31 TFCs were moderate performance.

## Linear relationships estimation

In this study, the linear relationships between technical effectiveness, bed utilization, and technical efficiency were investigated by applied two statistical models: correlation coefficient (r) and regression coefficient ($R^2$). We also assessed the impact of various factors on technical efficiency and death ratio and quantified their effects on these indicators. Table 4 showed that there was significant moderate positive correlation between technical efficiency and bed utilization rate (r = .431, n = 94, p < 0.001). However, there was a significant weak positive correlation of effectiveness to both technical efficiency and bed utilization, (r = .190, n = 94, p = .008) and (r = -.272, n = 94, p = .004).

The regression analysis results showed that the bed utilization rate interprets 7.4% of the change in the technical effectiveness score F (1,93) = 7.361, P = .008, $R^2$ = .074. However, there was no significant relationship between the technical effectiveness and the technical efficiency was statistically insignificant F (1,93) = 7.361, P = 0.428, $R^2$ = .008. Moreover, the regression analysis results showed that the technical effectiveness and the technical efficiency interpret 22.4% of the change in the bed utilization rate score F (1,93) = 13.097, P < 0.001, $R^2$ = .224. In addition, the bed utilization rate interprets

**Table 4. The significant correlation coefficient (r) and regression coefficient determination for the variables related to the bed utilization rate, efficiency, and effectiveness.**

| Dependent Variable (DV) | Independent Variable (IV) | Pearson's Correlation | | Linear Regression | | |
|---|---|---|---|---|---|---|
| | | r a | P value c | R-squared b | F (d.f.) | P value c |
| **Effectiveness** | Efficiency TA | .190 | 0.034 | .008 | 7.361 (1,93) | 0.428 |
| | Bed Utilization | .272 | 0.004 | .074 | 7.361 (1,93) | 0.008 |
| **Bed Utilization** | Effectiveness | .272 | 0.004 | .224 | 13.097 (1,93) | 0.000 |
| | Efficiency TA | .431 | 0.000 | | | |
| **Technical efficiency** | Effectiveness | .190 | 0.034 | .008 | 7.361 (1,93) | 0.428 |
| | Bed Utilization | .431 | 0.000 | .186 | 21.004 (1,93) | 0.000 |
| **Technical efficiency** | BTO | .586 | 0.000 | .805 | 123.90 (3,90) | 0.000 |
| | LOS | -.825 | 0.000 | | | |
| | OBR | .193 | 0.062 | | | |

18.6% of the change in the technical efficiency F (1,93) = 21.004, P < 0.001, $R^2$ = .186. In addition, the regression analysis model showed that the BTO, LOS and BOR interpret 81% of the change in the technical efficiency scores, F (3,90) = 123.90, P < 0.001, $R^2$ = .805. Finally, the regression analysis model showed that the number of staff and TFCs beds interpret 73% of the change in the technical efficiency scores, F (2,90) = 119.81, P < 0.001, $R^2$ = .725.

[a] r Correlation coefficient, [b] R-squared is a coefficient of determination P value at the significance level of 0.05. Finally, this study assessed the performance of the TFCs by measuring their technical effectiveness, efficiency and capacity of bed utilization based on quantitative way by using the input, output, and intermediate outcome data. It also analyzed the factors related to these indicators that were assessed by using the regression coefficient model. The findings indicated that TFCs had high technical performance with average scores of 82% and high effectiveness with an average score of 95%, and moderate technical efficiency with 73% average score, while the capacity of beds utilization was 63%. There were significant relationships among the three main indicators, but here it was low, ranging between 8% to 22%. The regression coefficient indicated that BTO and LOS interpret the change in the technical efficiency value, but no significant link with OBR.

## Discussion

Severe acute malnutrition impacts nearly 20 million children under five, mostly from the developing and poor countries [32,33]. The Nutrition Cluster has reported that the TFCs had admitted 17,613; 32,436; and 41,786 children with SAM/MC in the years 2020, 2021 and 2022, respectively. Moreover, the total number of children who lost their lives due to SAM/MC was 352, 422, and 802 children in 2020, 2021 and 2022, respectively [34]. This indicates that the nutritional status of children is deteriorating and is in its worst in Yemen. Tragically, it also reveals that 1,576 children have died from starvation in the span of three years. In this study, the results showed that 94 (67%) TFC enters supported by WHO treated 26,887 children with SAM/MC, 23,782 (90%) recovered, 944 (3.6%) default and 356 (1.3%) of them died. The overall these results were better than Sphere standards thresholds that state the recovered rate should be > 75%, defaulted rate < 15% and died ratio < 3% [35]. A systematic review and meta-analysis for 19 articles published between 2003–2019 found that the pooled average of recovery, defaulter, death and non-recovery rates were 70%, 10% 2% and 15% respectively [36]. In a recent article published in 2022, reported that the recovered rate was 68.72%, defaulter rate was 10.3% and death ratio was 4.32% [37].

Moreover, the study showed that the utilization of bed capacity was 48% for the BOR and 2.5 BTR and the average of stay in bed of children in the TFC centers was 7.1. Our observation close to those of Usaman et al., (2015) which reported a value close to our study results; BOR at 51.33% and LOS ranging between 3–30 days [38]. A study has also been carried out in Nigeria which reviewed the inpatient records for three teaching hospitals over a period of 7 years (2010–2016). The findings revealed that the BOR average was 42.14%, with a range of 22.62% to 62.37% [39].

Moreover, based on OECD database, the average of BOR in 6 developed countries was less than 60%, and 4.5 per 1,000 population of average of BTR [40]. In a study conducted in India, Sindhu et al., (2019) reviewed the bed utilization rate in 1948–2018 and showed that the BOR in 2018 was 74%, BTR was 39 and LOS was 10 days [41]. The study results show that the bed utilization rate is not much different from other countries. This includes Yemen, even though it is in a state of emergency. Secondly, the TFCs in Yemen are located and in various places across the country, including cities, rural and remote areas and so are easily accessible to low-income families. As a result, the BOR rate was affected by the low flow of children to those TFCs.

According to the Pabon model, the team's studies results have revealed that the performance scores of 79 (84%) TFCs were at acceptable levels. Only 15 FTCs (16%) had low performance (Zone I). The results also showed that all TFCs were effective, with 95%, with a very small standard deviation of 0.03% and 88% as minimum score. However, average of the efficiency was 73%, and bed utilization rate was 63%, the two indicators, scores were at moderate levels. Our findings are higher than what is found in a study conducted in Iran where the Pabon model showed that approximately half of hospitals

were inefficient (Zone I). Although the study found 20% of the hospitals were quite efficient, but it is better than our findings 8% are highly performance in Zone III [42].

In addition, the study's observations are aligned with three studies conducted in India, Jharkhand, Madhya Pradesh, and Gulbarga), and another two studies in Mali and Mauritania that concluded that inpatient therapeutic feeding programs were effective in the management of SAM children [43–47]. In studies conducted in, Mali, and in two regions in India (Mumbai and Eastern), they compared of therapeutic care offered by inpatient program and community program for SAM children. They found that the inpatient program was more effective than the community program, but the community program was low in cost compared to the inpatient program [48]. Nevertheless, the study's findings in some countries, like Ethiopia differ from the team's results. They found that the TFCs' effectiveness was low and below the Sphere standard threshold.

In Yemen, the World Health Organization has adopted a triple-pronged approach in dealing with the severe acute malnutrition. It involves securing financial support to provide free medical care for SAM children, and offering technical support to establish guidelines and protocols, intensive training, and continuous and close supervision and monitoring [49–51]. This was made possible by the generous assistance of a group of donors who hope to provide efficient and effective health care. Therefore, WHO is interested in achieving a balance between using resources wisely and providing effective and efficient medical care lifesaving services are essential for the service's survival and continuity. This is an important matter for Yemen because the prevalence of SAM/MC is growing every year, while the resources are limited, and the donations are decreasing. MoPHP and partners, particularly WHO, are therefore requested to devote more attention and effort to upgrade the efficiency and efficient bed utilization levels in the TFCs to be in line with the level of performance and effectiveness. As can be seen from the Table 4, we have examined the factors that influence efficiency. Hence, we have selected those factors that are not part of the allocated efficiency calculation by the DEA program. We found that the LOS and BTO have a strong correlation with the technical efficiency. We have put these two indicators and the BOR in the regression model, and so we found that they provide explanation for 81% of the variation in the efficiency scores.

The linear association and relationships between the indicators and between their variables were assessed by using correlation coefficient and regression coefficient models. Table 4 showed that there is a moderate positive correlation between efficiency and bed utilization rate (r = .431), and between effectiveness and bed utilization rate (r = .272). The correlation between effectiveness and efficiency was positive week (r = .190). Moreover, the regression coefficient model appeared that there is non-statistically significant relationship between effectiveness and efficiency ($R^2$ = .008, P value > 0.05), while the relationship between effectiveness and bed utilization was statistically significant (P value = 0.008), but the bed utilization interprets only 7.4% of changes in effectiveness scores. The relationship between efficiency and bed utilization was statistically significant ($R^2$ = .224, P value < 0.001), i.e., the bed utilization rate interprets 22.4% of changes in the efficiency scores.

Our studies are consistent with a study evaluated the health system performance in 32 European countries from 2011 to 2014. The results indicated that the efficiency and effectiveness levels uncorrelated, they had different trends and patterns over time. They highlighted that the two indicators may not have a trade-off relationship between them [21]. In another study had been conducted in Italy, the study aimed to assess magnitude of trade-offs between efficiency and effectiveness (mortality and readmission) in the years 2008–2011. The results indicated that there is negative association between efficiency against both mortality and readmission [52]. However, our findings were not consistent with some other research. Bartuševičienė and Šakalytė (2013), conducted a general scientific review, the reviewing aimed to study the differences and proximities between effectiveness and efficiency. The results indicated that the effectiveness and efficiency influence each other, and they should be correlated, but it is not necessarily to be related [53]. A systematic review examined how efficiency and hospital size were related. The result did not find statistically significant association between the two variables [54]. Some studies have shown a significant relationship between efficiency and effectiveness, and how they affect each other. However, another some studies have suggested that there may not be a trade-off between the two

indicators, meaning that they may be correlated, but not necessarily related. This is what we found in our study, where we found a moderate correlation, but no significant dependency between them.

This study had several limitations, which are: first limitation, the study only covered 94 out of 141 TFCs that were supported by WHO, because the data from the other centers, which had different partners, was inaccessible. This could affect the external validity, but we tried to minimize this threat by including all the accessible TFCs, which represented 67% of the total TFCs. However, this procedure was necessary to avoid the potential of selection bias, thus minimizing the threat to the internal validity by selecting only the TFCs that are supported by WHO. Second limitation, a possible risk to the internal validity of this study is the variation in the type and location of TFCs, which may affect their overall performance. Hence this study tries to reduce this bias by selecting only TFCs that are supported by WHO, which provide homogenous services and follow consistent policies and procedures that are applied in all TFC centers. Third limitation, this study may not capture the broader aspects of quality, value added, employee satisfaction, and the output interaction with the social and economic environment. Therefore, it is recommended to examine effectiveness from both qualitative and quantitative perspectives and integrate them in one model in future studies. Fourth limitation. due to the weak or moderate relationships between the main variables, the stepwise method was applied to enhance the model, and systematically included and excluded the variables, ultimately improving the model's predictive accuracy. A final limitation of this study is that it only uses data based on one-year practices, which may not reflect the long-term effects of the TFCs policy that aims to improve the performance from both efficiency and effectiveness perspectives. Therefore, it is suggested to use a longer data span in future studies to capture the temporal changes in the performance of TFCs.

## Conclusion

The results indicated that the TFCs performed well performed well overall, even exceeding Sphere standards thresholds for the intermediate outcomes indicators. However, there is still a need to improve efficiency and bed utilization to meet the standards. The study identified to a positive correlation between effectiveness and efficiency, although no direct causal relationship was established. Therefore, to enhance performance further, it is crucial for the Ministry of Health and its partners to optimize resource utilization and strengthen coordination.

## Supporting information

**Table S1–S2. TFC centers database and analysis data.**
(RAR)

## Acknowledgments

The data for this study was gathered by the statisticians at the TFC centers and verified by the M&E officer of the WHO organization. We extend our sincere thanks to them for kindly sharing this data with us. Additionally, we would like to thank Associate Professor Mohammad Sharfuddin, English department, Sana'a university for his invaluable assistance with proofreading and editing of the final manuscript draft.

## Author contributions

**Conceptualization:** Mohammed Rajamanar.

**Data curation:** Mahammad Al Mansour, Latifah Ali.

**Formal analysis:** Ahmed Alhidary, Ali Al-Waleedi, Ali AL-Mudwahi, Omar Faisal, Eshrak Al-Falahi.

**Investigation:** Latifah Ali.

**Methodology:** Ahmed Alhidary, Ali Al-Waleedi.

**Project administration:** Pesigan Arturo, Ferima Coulibaly-Zerbo, Eshrak Al-Falahi, Latifah Ali.

**Resources:** Mohammed Rajamanar, Eshrak Al-Falahi.

**Supervision:** Pesigan Arturo, Ferima Coulibaly-Zerbo, Omar Faisal.

**Validation:** Ezechiel Bisalinkumi.

**Visualization:** Mahammad Al Mansour.

**Writing – original draft:** Ahmed Alhidary, Ezechiel Bisalinkumi.

**Writing – review & editing:** Pesigan Arturo, Ferima Coulibaly-Zerbo, Ali AL-Mudwahi.

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
