## [Decision Letter · Decision Letter 0]

7 Aug 2024

PONE-D-24-16119Evaluating the Performance of Therapeutic Feeding Centers through Efficiency, Effectiveness and Utilization of Bed Capacity, and Assessing Related FactorsPLOS ONE

Dear Dr. Alhidary,

Thank you for submitting your manuscript to PLOS ONE. After careful consideration, we feel that it has merit but does not fully meet PLOS ONE’s publication criteria as it currently stands. Therefore, we invite you to submit a revised version of the manuscript that addresses the points raised during the review process.

Your article needs major revision to be published. Please make any changes requested by the reviewer, paying attention to data availability. Also, consider whether you need to update your conflict of interest statement based on the non-financial competing interest “Membership on a government or other advisory board.” https://journals.plos.org/plosone/s/competing-interests

Please submit your revised manuscript by Sep 21 2024 11:59PM. If you will need more time than this to complete your revisions, please reply to this message or contact the journal office at plosone@plos.org . Please include the following items when submitting your revised manuscript:

We look forward to receiving your revised manuscript.

Kind regards,

Zenewton André da Silva Gama, Ph.D.

Academic Editor

PLOS ONE

Journal Requirements: 

"The authors have declared that no competing exist"

4. In the online submission form, you indicated that [All relevant data are within the manuscript and its supporting information files. Should additional details be required,we maintain a database that can be accessed upon request.]. 

5. Please amend your authorship list in your manuscript file to include author Dr. Omar FAISAL. 

6. Please ensure that you refer to Figure 1 in your text as, if accepted, production will need this reference to link the reader to the figure.

Reviewers' comments:

Reviewer's Responses to Questions

**Comments to the Author**

1. Is the manuscript technically sound, and do the data support the conclusions?

Reviewer #1: Partly

2. Has the statistical analysis been performed appropriately and rigorously? 

Reviewer #1: No

3. Have the authors made all data underlying the findings in their manuscript fully available?

Reviewer #1: Yes

4. Is the manuscript presented in an intelligible fashion and written in standard English?

Reviewer #1: No

5. Review Comments to the Author

Reviewer #1: “Evaluating the Performance of Therapeutic Feeding Centers through Efficiency, Effectiveness and Utilization of Bed Capacity, and Assessing Related Factors” for Plos One.

The objective of this study was to examine the efficiency, effectiveness, bed utilization capacity, and factors associated with the performance of Therapeutic Feeding Centers (TFC). An integrated nutrition service of this nature is crucial to addressing undernutrition and rehabilitating children suffering from severe malnutrition. Therefore, it is imperative to assess the quality of health services in order to provide guidance to public sponsors and donors.

I believe that certain aspects require thorough scrutiny:

All abbreviation terms must have been previously described in the abstract and in the body text.

Introduction: There is new data about children with severe acute malnutrition, please update reference 3.

Introduction: Please clarify the information about “reporting rate” from OTP and TFC.

Introduction: What is TVC?

Introduction: The authors affirm that “there is no study that has investigated performance evaluation of inpatient programs in a quantitative manner, either locally or regionally”. Please clarify this point, since in the Discussion section the authors mention many other studies in the same sense.

Introduction: Explain and support the statement “The research activities are not of great importance for the countries that have high prevalence of SAM/MC”.

Introduction: Why do the authors not consider “bed utilization” as a variable to evaluate efficiency, as proposed by Pabon Lasso model?

Methods: Which studies are made to select the “six intermediate outcome variables to determine the average for the effectiveness score”? Please give details about the population of these studies.

Methods: Please describe the cure indicator. It was based on clinical overcoming, or achieving and maintaining any anthophometric indice, or any point in the WHO growth chart reference?

Methods: Why do the authors choose the stepwise method for regression analysis, even after a weak correlation? This point could be added to the study limitations.

Methods: The “Ethical Considerations” shown in the body text differ from those described in “Ethics Statement” during the submission process. Please provide the website (or other) where the public data can be found, as you inform us in "This study utilized publicly available secondary data".

Results: The table titles must be revised to reflect the content. Legends could be added to describe abbreviations. It is not necessary to write the same information that could be found on the table.

Results: Explain in the regression analysis which variables were modeled as predictors. This point must be retaken in the Discussion section.

Discussion: Why are there data from admitted and died children two times in 2022?

Discussion: Please explain the statement “It also indicates that the most tragic event mash?”

Discussion: What does it mean “(...) nineteen articles reviewed the articles from 2003 to 2019(...)?”

Discussion: In the cited reference 38, 68.72% referes to “recovery” children not to “cure rate” as written.

Discussion: I agree with the authors when they affirm that “this study had several limitations”, but maybe with a clearly statistical approach to the variables analyzed, the number of TFC covered could be sufficient to reflect the external and internal validity. I suggest being aware of the hypotheses, correctly describing the variables, and clearly explaining the results.

Conclusion: In this section, avoid repeating the numerical results and solely answer your objectives explaining the importance of information for the maintenance or improvement of the TFC.

The textual grammar, coherence, and cohesion must be reviewed to improve the quality of the paper.

Aspects like standard terms, numeric formatation, and signal uses must be revised carefully.

6. PLOS authors have the option to publish the peer review history of their article (what does this mean? ). If published, this will include your full peer review and any attached files.

**Do you want your identity to be public for this peer review?** For information about this choice, including consent withdrawal, please see our Privacy Policy .

Reviewer #1: No

---

## [Author Response · Author response to Decision Letter 1]

7 Sep 2024

Dear Sir/Madam/Reviewer, greetings from Yemen

Thank you for your thorough review of our manuscript titled " Performance Evaluation of Therapeutic Feeding Centers through Efficiency, Effectiveness and Utilization of Bed Capacity: A retrospective quantitative study" We sincerely appreciate the time and effort you have dedicated to providing us with your valuable and insightful comments, which have significantly contributed to enhancing the quality of our work.

We have carefully considered each of the comments and suggestions provided and have made the necessary revisions to the manuscript accordingly. We hope that these changes meet your expectations and improve the clarity and robustness of our manuscript.

Thank you once again for your valuable feedback. Below, we provide a detailed, point-by-point response to each comment, with our responses highlighted in yellow for ease of reference.

Sincerely,

Authors

Journal Requirements:

The authors, well noted, done.

The authors will submit the article using the PLOS LaTeX template after completing the final draft and review.

"The authors have declared that no competing exist"

The authors, well noted, done.

4. In the online submission form, you indicated that [All relevant data are within the manuscript and its supporting information files. Should additional details be required, we maintain a database that can be accessed upon request.].

All PLOS journals now require all data underlying the findings described in their manuscript to be freely available to other researchers, 1. In a public repository, 2. Within the manuscript itself, or 3. Uploaded as supplementary information.

The authors, well noted, done.

5. Please amend your authorship list in your manuscript file to include author Dr. Omar FAISAL.

The authors, well noted, done.

6. Please ensure that you refer to Figure 1 in your text as, if accepted, production will need this reference to link the reader to the figure.

Reviewers' comments:

Reviewer's Responses to Questions

The authors, well noted, done.

Comments to the Author

1. Is the manuscript technically sound, and do the data support the conclusions?

Reviewer #1: Partly

2. Has the statistical analysis been performed appropriately and rigorously?

Reviewer #1: No

3. Have the authors made all data underlying the findings in their manuscript fully available?

Reviewer #1: Yes

4. Is the manuscript presented in an intelligible fashion and written in standard English?

Reviewer #1: No

5. Review Comments to the Author

Reviewer #1: “Evaluating the Performance of Therapeutic Feeding Centers through Efficiency, Effectiveness and Utilization of Bed Capacity, and Assessing Related Factors” for Plos One.

The objective of this study was to examine the efficiency, effectiveness, bed utilization capacity, and factors associated with the performance of Therapeutic Feeding Centers (TFC). An integrated nutrition service of this nature is crucial to addressing undernutrition and rehabilitating children suffering from severe malnutrition. Therefore, it is imperative to assess the quality of health services in order to provide guidance to public sponsors and donors.

I believe that certain aspects require thorough scrutiny: All abbreviation terms must have been previously described in the abstract and in the body text.

Done,

Introduction: There is new data about children with severe acute malnutrition, please update reference.

Done,

Introduction: Please clarify the information about “reporting rate” from OTP and TFC.

The Outpatient Therapeutic Program (OTP) is a treatment approach designed for managing children aged 6-59 months with moderate acute malnutrition (MAM) or uncomplicated severe acute malnutrition (SAM). OTP provides ready-to-use therapeutic food (RUTF) and standard medical care, serving both children currently in the community and those who have recovered from severe acute malnutrition and been discharged from inpatient Therapeutic Feeding Centers (TFCs).

Therapeutic Feeding Centers (TFCs) are specialized inpatient facilities designed to provide intensive nutritional and medical care to children suffering from severe acute malnutrition with complications. These centers offer a comprehensive approach to treatment, including specialized feeding regimens, medical interventions, and monitoring to address the complex needs of severely malnourished children.

Introduction: What is TVC?

The authors: Sorry it is typographical error, the correct was mandate. It is TFC

Introduction: The authors affirm that “there is no study that has investigated performance evaluation of inpatient programs in a quantitative manner, either locally or regionally”. Please clarify this point, since in the Discussion section the authors mention many other studies in the same sense.

The authors: The authors confirm that there is a significant gap in studies evaluating the performance of therapeutic feeding centers (TFCs) and inpatient services in the region. The Arab world ranks among the lowest globally in scientific research. Despite extensive reviews of research databases and university libraries, we did not find any studies conducted in Yemen or the broader Arab region on this topic. While the discussion section references several studies, these were conducted outside the region, such as in China and New Zealand (see references 20 to 22). Therefore, this article represents the first effort to evaluate inpatient unit performance in Yemen and will be one of the rare comprehensive evaluations conducted at this level in our region.

Introduction: Explain and support the statement “The research activities are not of great importance for the countries that have high prevalence of SAM/MC”.

The authors: Countries with a high prevalence of Severe Acute Malnutrition (SAM) and Moderate Acute Malnutrition (MAM) often face critical, life-threatening situations that demand immediate action. Consequently, these countries are compelled to prioritize toward relief efforts rather than research initiatives. Due to limited resources, urgent needs, unstable conditions, and inadequate infrastructure and capacity, these countries rank among the lowest globally in terms of scientific research output.

Introduction: Why do the authors not consider “bed utilization” as a variable to evaluate efficiency, as proposed by Pabon Lasso model?

You are correct; bed utilization is a key indicator of efficiency. In calculating the overall TFC performance score, we assigned equal weights to technical efficiency (25%) and bed utilization (25%), while technical effectiveness was given a weight of 50%. This means that bed utilization and technical efficiency together accounted for half of the total score, compared to effectiveness.

Moreover, bed utilization was assessed using the variables Occupancy Bed Rate (OBR), Bed Turnover (BTO), and Length of Stay (LOS). While some studies classify LOS as a measure of efficiency, others consider it a measure of effectiveness. The length of stay (LoS) in hospitals is a key metric used to estimate both the efficiency and effectiveness of inpatient care. LoS can indicate how well a hospital is managing its resources and patient flow. Shorter stays often suggest efficient use of resources and effective care, while longer stays might indicate complications or inefficiencies.

Additionally, the technical efficiency was calculated using the Data Envelopment Analysis Program (DEAP), which assesses the relative efficiency of units that utilize the same inputs to produce the same outputs. While the indicators such as OBR, BTO, and LOS are component indicators and, therefore, are not directly applicable for inclusion in the model.

Methods: Which studies are made to select the “six intermediate outcome variables to determine the average for the effectiveness score”? Please give details about the population of these studies.

The authors: We based on several studies; the following is example.

A systematic review conducted in 2020, with title “Assessing hospital performance indicators. What dimensions? Evidence from an umbrella review”. This study in the Table 2, determined the variables that measure the effectiveness. The population study in this study is hospitals.

The study has been conducted in Ethiopia with title “Cost effectiveness of community-based and inpatient therapeutic feeding programs to treat severe acute malnutrition in Ethiopia”. Please refer to the section “Measuring Effectiveness” The population study in this study is TFC programs.

The study has been conducted in Virginia with title “Patient care effectiveness and financial outcomes of hospital physician contracting emphasis” Please refer to the section “Dependent measures, summary table 1”. In this article, the authors used length of stay in bed, readmissions, and mortality as indicators to assess the inpatient effectiveness. The population study in this study is inpatient departments.

Methods: Please describe the cure indicator. It was based on clinical overcoming, or achieving and maintaining any anthropometric indices, or any point in the WHO growth chart reference?

Based on the WHO-Yemen country guideline; To describe the "cure indicator" used in Therapeutic Feeding Centers (TFCs), a combination of clinical and anthropometric criteria is typically applied to determine if a child has recovered from Severe Acute Malnutrition (SAM). 1) Clinical Improvement: This involves resolving medical complications associated with SAM, such as infections, dehydration, and other life-threatening conditions. Additional criteria for clinical recovery include the child's appetite returning (as confirmed by passing the appetite test), reduction of severe bilateral pitting edema for two consecutive weeks, and the child appearing clinically well and alert. For children admitted with both bilateral pitting edema and severe wasting, recovery is indicated when the edema has completely resolved. 2) Anthropometric Indices: The cure indicator also relies on specific anthropometric measurements, such as maintaining a weight-for-length/height z-score (WHZ) above -2 for at least two consecutive days or achieving a mid-upper arm circumference (MUAC) of ≥125 mm.

Methods: Why do the authors choose the stepwise method for regression analysis, even after a weak correlation? This point could be added to the study limitations.

The authors: We added this in the limitation section, many thanks for this valuable comment

The relationships between effectiveness and efficiency, as well as between effectiveness and bed utilization, are weak, while the correlation between efficiency and bed utilization is moderate. To refine the model, the stepwise method was used, which systematically adds or removes variables based on their statistical significance. This approach helps identify the most relevant predictors, even when overall correlations are minimal, and improves the model's predictive accuracy by selecting the optimal combination of variables.

Methods: The “Ethical Considerations” shown in the body text differ from those described in “Ethics Statement” during the submission process. Please provide the website (or other) where the public data can be found, as you inform us in "This study utilized publicly available secondary data".

Well noted, done.

Results: The table titles must be revised to reflect the content. Legends could be added to describe abbreviations. It is not necessary to write the same information that could be found on the table.

Well noted, done.

Results: Explain in the regression analysis which variables were modeled as predictors. This point must be retaken in the Discussion section.

The authors: The study utilized secondary data obtained from the WHO’s monitoring and evaluation unit, adhering to all relevant ethical guidelines and regulations. The data were anonymized to prevent the disclosure of any personally identifiable information. The Ministry of Health granted permission to use the data, and four ministry employees co-authored the article. Efforts were made to ensure data accuracy, and all potential risks associated with data use were assessed to minimize harm to individuals or groups.

Discussion: Why are there data from admitted and died children two times in 2022?

The authors: Apologies for the typographical error; it has been corrected in the current manuscript draft.

Discussion: Please explain the statement “It also indicates that the most tragic event mash?”

The authors: Apologies for the typographical error; it has been corrected in the current manuscript draft.

Discussion: What does it mean “(...) nineteen articles reviewed the articles from 2003 to 2019(...)?”

The authors: Apologies for this error; it has been corrected in the current manuscript draft.

Discussion: In the cited reference 38, 68.72% referes to “recovery” children not to “cure rate” as written.

The authors: we explained that “The cure indicator is summation of both cure and recovery cases.”

Discussion: I agree with the authors when they affirm that “this study had several limitations”, but maybe with a clearly statistical approach to the

---

## [Editor Report · Decision Letter 1]

13 Dec 2024

Performance Evaluation of Therapeutic Feeding Centers through Efficiency, Effectiveness and Utilization of Bed Capacity: A retrospective quantitative study

PONE-D-24-16119R1

Dear Dr. Alhidary,

We’re pleased to inform you that your manuscript has been judged scientifically suitable for publication and will be formally accepted for publication once it meets all outstanding technical requirements.

Kind regards,

Zenewton André da Silva Gama, Ph.D.

Academic Editor

PLOS ONE

Additional Editor Comments (optional):

The authors have addressed all reviewer comments, made necessary clarifications and corrections, and revised the manuscript for greater accuracy and coherence.
---

## [Editor Report · Acceptance letter]

PONE-D-24-16119R1

PLOS ONE

Dear Dr. Alhidary,

I'm pleased to inform you that your manuscript has been deemed suitable for publication in PLOS ONE. Congratulations! Your manuscript is now being handed over to our production team.

Kind regards,

on behalf of

Prof. Dr. Zenewton André da Silva Gama

Academic Editor

PLOS ONE